# The Impact of a Terminal High Altitude Area Defense Incident on Tourism Risk Perception and Attitude Change of Chinese Tourists Traveling to South Korea

**Hao Zhang** [1,†]**, Taeyoung Cho** [2,†] **and Huanjiong Wang** [1,]***

1  Key Laboratory of Land Surface Pattern and Simulation, Institute of Geographic Sciences and Natural Resources Research, Chinese Academy of Sciences, 11A, Datun Road, Chaoyang District, Beijing 100101, China; zhanghao@igsnrr.ac.cn

2  Department of Hotel Tourism Management, Dongguk University, 123, Dongdae-ro, Gyeongju-si, Gyeongsangbuk-do 38066, Korea; cty0629@dongguk.ac.kr

*  Correspondence: wanghj@igsnrr.ac.cn

†  The first two authors contributed equally to this work.

**Abstract:** The crisis event, which is one of the factors that occur most frequently, affects the sustainable development of tourism. Aiming to investigate the impact of a terminal high altitude area defense (THAAD) incident on tourism risk perception, destination image, attitude change and behavior intention of Chinese citizens planning to travel to South Korea, a questionnaire survey was conducted among 1000 Chinese citizens. By using descriptive statistics, factor analysis, and a structural equation model on the survey data, the results showed that (1) tourist destination risk perception exerted a negative effect on tourism image, culture image, and stability image of the destination; (2) the risk perception of tourists on tourism destinations would lead to a change in tourism attitudes, as Chinese citizens who planned to travel to South Korea considered changing their tourist destinations due to the THAAD incident; and (3) the increased risk perception of tourists on destinations had negative effects on behavior intention. Chinese citizens' intention to travel to Korea was reduced after the THAAD incident. This study is of great importance for the sustainable development of tourism between China and Korea and provided a reference for improving risk management strategy for a tourism destination.

**Keywords:** tourism risk perception; destination image; attitude change; tourist behavioral intention; THAAD incident

## 1. Introduction

Along with the increasing number of reports on exogenous risks such as terrorism and natural disasters, there is an exponential increment in the number of risk researches in tourism [1]. Tourism risk is when tourists, in considering their travel behavior, perceived negative results that may occur [2]. When tourists choose a tourist destination, they would make a judgment of the security and uncertainty of tourism activities, which is a process called tourism risk perception [3]. Tourists' risk perception often has five to seven dimensions, namely artificial risk, financial risk, service quality risk, psychosocial risk, the risk of natural disasters and accidents, food safety issues, and weather [4,5]. The assessment of tourism risk perception is of significance for making tourism behavioral decisions and destination security management. Acceptable risk has a positive feedback effect on tourists' decisions, and unacceptable risk has a guiding role in the risk of tourist destination control [6].

Tourism risk perception is a negative factor for the healthy development of tourism destinations and seriously affects tourism destination's attractiveness [7]. For example, on August 8, 2017, coincided

with the peak tourist season, a magnitude 7.0 earthquake in Jiuzhaigou, China, severely damaged the natural landscape in the Jiuzhaigou scenic area. After this event, because the tourists perceived the risk of natural disasters for this tourism destination, the number of tourists reduced sharply [8]. Tourism risk perception is also closely related to travel attitude change. Commonly in travel decision-making, tourists will try to reduce the security risks encountered in tourism activities [9]. For example, after the "911" incident in 2001, tourists who originally planned to go to the United States began to find a relatively safe place, and changed their tourist destinations or completely canceled their tourism activities [10]. Tourist behavioral intention, a kind of attitude of tourists towards tourism destinations, is also impacted by tourism risk perception. Especially for cross-border tourism activities, the consumer hostility caused by various conflicts between the two countries affects consumers' willingness to travel [11]. For example, in September 2012, the Japanese government purchased three of the Diaoyu islands from their private owner, prompting large-scale protests in China because the Diaoyu Islands concerns a territorial dispute between Japan and China. This event has reduced the willingness of Chinese tourists to travel to Japan, leading to large-scale canceling travel to Japan, and most Chinese travel agencies have suspended Japanese tourism projects [12]. Therefore, tourism risk perception has a potential impact on destination image, attitude change, and tourism attitude change.

Tourism is a sensitive and fragile industry that is very vulnerable to the impact of various crisis events and emergencies. Crisis events can affect the tourism risk perception, and thus alter the confidence of tourists in destinations and travel plan of tourists, leading to significant changes in the market structure of tourist destinations and source areas [13]. In recent years, various global crises have shown an upward trend. A series of terrorist incidents (such as the "911" incident in the United States, and the terrorist attacks in Paris, France), and various natural disasters (such as the Indonesian tsunami and the Wenchuan earthquake in China) have brought a considerable impact the tourism industry. For instance, after the terrorist attack in Paris in November 2015, major travel companies and agencies have urgently canceled tourism activities in Paris, severely affecting the French economy [14]. Overall, limited studies focused on tourism attitudes change after such crisis events. This study focused on a terminal high altitude area defense (THAAD) incident between Korea and China since South Korea's decision to deploy THAAD to protect itself against North Korea, which has caused backlash and retaliation measures from China. This incident may affect the Chinese citizens' tourism risk perception, attitude change, and behavioral intention of traveling to South Korea.

On July 8, 2016, the South Korean government announced that it would deploy THAAD at the US military base in South Korea. This event affected the enthusiasm of Chinese tourists to South Korea. Before the THAAD incident, because of the popularity of Korean pop culture, the medical beauty tourism industry, and the fashion elements such as cosmetics and apparel, more and more Chinese citizens traveled to South Korea on weekends and holidays. South Korea is the most popular overseas destination for Chinese people [15]. According to data from the Korea Tourism Organization, in 2013, 4.32 million Chinese citizens traveled to South Korea, and China became the country with the largest number of people among Korean inbound tourists [16]. After the THAAD incident, the relations between the two countries significantly deteriorated. China imposed an unofficial boycott on South Korea. On July 3, 2017, the National Tourism Administration of China issued "Travel Risk Tips to Korea" to the public, and the enthusiasm of Chinese citizens to travel to South Korea declined sharply. Therefore, the THAAD incident provided a chance to explore Chinese citizens' attitude change and behavioral intention to travel to Korea under a major crisis. This study conducted a questionnaire survey among 1000 Chinese citizens about their intention to travel to Koreas after the THAAD incident and applied descriptive statistics, factor analysis, and a structural equation model on the survey data. Our objective is to investigate the impact of the THAAD incident on tourism risk perception, destination image, attitude change, and behavior intention of Chinese citizens planning to travel to South Korea.

This paper starts with the literature review, research hypotheses, and a model proposed. It is followed by the research design about the method of data collection and statistical analysis.

Subsequently, the result of factor analysis, convergent and discriminant validity, and the structural equation model are shown. Finally, it is concluded with a summary of the comparison with previous studies and practical recommendations for tourism management departments and organizations.

## 2. Literature Review, Research Hypotheses, and Model

### 2.1. Literature Review

(1). *Tourism Risk Perception*

Since tourists' travel decision largely depends on the cognition of the destination environment, especially the safety environment, tourist destination marketing decision-makers need to quantify the destination's security. However, Shahid [17] proposed that security cannot be quantified, and it is necessary to link security with risk. In Moutinho [18], tourism risk perception was divided into five categories: functional risk, physical risk, economic risk, social risk, and psychological risk. Fuchs [19] investigated the tourism risk perception of foreign tourists who visited Israel and identified six target risk perception factors: human risk, finance, service quality, psychosocial, natural accidents and car accidents, food safety issues, and weather. Overall, risk perception referred to the individual's perception and feeling of the objective risk that exists in the outside world and emphasized the effect of individual experience acquired through direct judgment and subjective perception on cognition [20]. Tourism risk refers to the possibility of tourists suffering from various misfortunes in tourism activities [21]. In recent years, scholars regarded that tourism risk perception is (1) tourist's subjective judgment and cognition; (2) tourists' perception of uncertainty and consequences of risk events; (3) involving many aspects of tourism activities; and (4) easy to be affected by many factors [17–21]. In this study, the tourism risk perception refers to the subjective cognition and feeling of tourism risk by tourists under the influence of external information and self-factors in tourism activities.

(2). *Destination Image*

With the competition between tourist destinations becoming increasingly fierce, the destination image, as the major factor in attracting tourists and creating local identity, has become the core source of the competitive advantage of tourist destinations [22]. The destination image perception is an important factor for tourists to decide whether to visit the destination. Therefore, how to accurately grasp the destination image perception of tourists and potential tourists has become a top priority in tourism research. At present, the formation factors of tourism destination image included the tourist destination perception and emotional factors [23,24]. The tourist destination perception refers to the tangible factor (e.g., cultural heritage, natural scenery), while the emotional factor refers to the evaluation of the tourist destination based on the tourists' personal values and the emotion caused by the tourist destination [25]. Bi [26] pointed out that the destination image is the overall cognition and subjective impression of travel, culture, stability, and instability about the destination, which is the definition adopted in this study.

(3). *Tourism Attitude Change and Tourist Behavioral Intention*

Tourism attitude is the tourists' evaluation and behavioral tendency, reflecting tourists' real feelings about tourism activities. Attitude change mainly refers to the change in altitude after the tourists receive some information or opinions. Tourism attitude is an important factor affecting tourists' intention to travel and directly determines tourists' travel behavior [27]. Gartner [28] considered that tourist intention is a kind of attitude of tourists towards tourism destinations. Oliver and Swan [29] pointed out that tourist intention is the most direct predictor of many factors affecting tourism behavior, and it is also an attitude and tendency of tourists towards to future tourism behavior. Most studies believed that attitude, previous experience, satisfaction, perceived value, image, motivation, and other factors are the main influencing factors of behavior intention [30,31]. Thus, the tourist behavioral intention is easily affected by a variety of factors, such as the impression of a tourism destination and

the perception of tourism risk, which all have a significant impact on tourist behavior. Especially for cross-border tourism activities, not only the consumer hostility caused by various conflicts between the two countries affects consumers' willingness to travel, the hostility generated during population flow and migration will also greatly reduce tourist behavioral intention [11]. In this study, the tourist behavior intention refers to the tourist's willingness to travel, plan to travel, or promise to travel in the future.

*2.2. Research Hypotheses*

(1). *Tourism Risk Perception and Destination Image*

Tourism safety not only affects the healthy development and attractiveness of tourism destinations but also has a considerable effect on the whole society and economic development [7]. Li et al. [32] showed that tourist risk perception has a negative effect on the number of tourists for tourist destinations. Chew and Jahar [33] analyzed the degree of Japanese tourists' risk perception and the willingness to revisit after the disaster and found that the tourists' social-psychological risk and financial risk had a significant impact on tourism destination image. Based on these studies, we proposed hypothesis 1.

**H1:** Tourism risk perception has a significant impact on destination image.

**H1-1:** Tourism risk perception has a significant negative impact on the tourism image.

**H1-2:** Tourism risk perception has a significant negative impact on the cultural image.

**H1-3:** Tourism risk perception has a significant negative impact on the stability image.

**H1-4:** Tourism risk perception has a significant positive impact on the instability image.

(2). *Tourism Risk Perception and Travel Attitude Change*

Tourism risk perception is closely related to travel attitude change. In travel decision-making, tourists will try to reduce the security risks encountered in tourism activities. If the tourists are not satisfied with tourist activities or feel less secure in tourism destinations, they would decide to change the original decision and attitude. Sönmez and Graefe [34] found that the level of tourism destination risk perception exerted a significantly negative impact on the possibility of choosing to travel there. Kozak [35] also found that if tourists did not feel secure enough, they would change or cancel their travel plans directly. There were obvious differences in risk perception between first-time and revisit tourists, but they all changed their attitude when perceiving risk [4]. Wang [36] investigated the tourism risk perception of senior citizens and found that tourism risk perception has a positive impact on tourism attitude change. Based on these studies, we proposed hypothesis 2.

**H2:** Tourism risk perception has a significant positive impact on tourism attitude change.

(3). *Tourism Risk Perception and Tourist Behavioral Intention*

Although the level of tourism risk does not always correlate negatively with tourists' intention, the safety requirements of tourists could affect tourist behavioral intention. Therefore, risk perception is the basis of tourism decision-making. According to Cater [37], tourists' subjective risk perception had a greater impact on their travel behavioral intention. The result of Richard [38] found that tourism safety risk perception has an impact on tourist behavior intention. Moufakkir [11] pointed out that consumer hostility caused by various political conflicts between two countries would affect consumers' travel behavior intention, and this negative emotion had a significant negative effect on consumers' willingness to travel. Based on these studies, we proposed hypothesis 3.

**H3:** Tourism risk perception has a significant negative impact on tourist behavioral intention.

### 2.3. Research Model

Based on the above literature review and research hypotheses, this study constructs a structural equation model about tourism risk perception, destination image, tourism attitude change, and behavior intention. All three hypotheses are summarized in Figure 1.

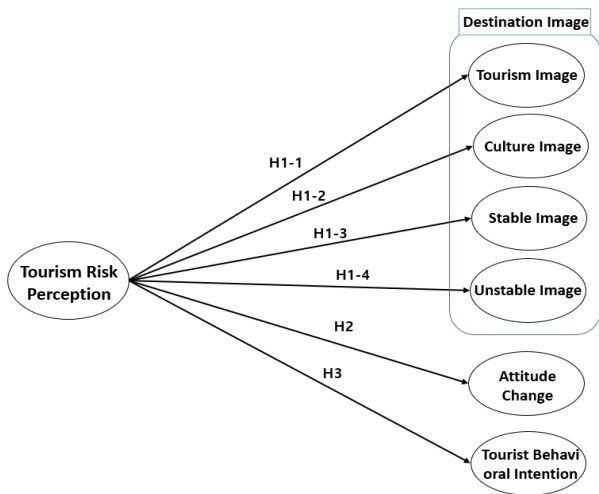

**Figure 1.** The proposed model describing the relationship between tourism risk perception, destination image, attitude change, and behavioral intention.

## 3. Research Design

### 3.1. Questionnaire Design and Measurement Scale Composition

The questionnaire consisted of tourism risk perception, destination image, tourism attitude change, tourist behavioral intention, and demographic characteristics of the population (Table 1). The measurement index of tourism risk perception [33], destination image [26], tourism attitude change, [4] and tourism behavior intention [34] were obtained from corresponding references. The indicators of the measurement scale were all evaluated using the Likert 5-level scale. Respondents chose among "1" = completely disagree, "2" = disagree, "3" = neither disagree or agree, "4" = agree, and "5" = completely agree. The questionnaire also included the demographic characteristics of the respondents, such as gender, marital status, age, education level, average monthly income, occupation, etc.

### 3.2. Questionnaire Survey and Data Collection

Through consulting and interviewing staffs of international travel agencies, we understood the characteristics of current and potential markets about Chinese citizens traveling to South Korea. Subsequently, we conducted a questionnaire survey on the Chinese citizens who have a certain economic ability and leisure time, as well as other objective conditions concerning travel to Korea, and the tourists who had been to Korea. In order to increase the coverage of the questionnaire samples, questionnaires were distributed to Chinese citizens based on the customer resource databases in international travel agencies. Furthermore, investigators who had professionally trained were commissioned to conduct on-street questionnaires in 34 provinces, municipalities, and autonomous regions of China. After selecting eligible respondents through face to face interviews, the investigators distributed the questionnaire. From March 1, 2017, to November 30, 2017, 1000 questionnaires were distributed, and 986 questionnaires were recovered. Excluding wrong filled or unfinished questionnaires, a total of 970 questionnaire samples were used in the following analysis.

**Table 1.** Questionnaire Design.

| Values | Measurement Items |
|---|---|
| Tourism Risk Perception | TRP1. The THAAD incident would bring instability and insecurity to Korean society.<br>TRP2. The THAAD incident would lead to an unsafe political situation in South Korea and even caused war.<br>TRP3. There would be conflicts and accidents during the tour due to the THAAD Incident.<br>TRP4. The negative news and report on the THAAD incident would bring risks and insecurity. |
| Destination Image (Tourism Image) | DIT1. South Korea is a country with developed tourism.<br>DIT2. South Korea has a variety of festivals and various tourism activities.<br>DIT3. South Korea's night-time tourism activities and projects are colorful.<br>DIT4. The entertainment life in South Korea is wonderful. |
| Destination Image (Culture Image) | DIC1. South Korea has abundant cultural heritage resources<br>DIC2. South Korea has its own culture and tradition, and they are well preserved. |
| Destination Image (Stable Image) | DIS1. South Korea is a very attractive country.<br>DIS2. South Korea is a country with humanism.<br>DIS3. The city and tourist areas in South Korea are clean and comfortable.<br>DIS4. Korean society is developed, and the transportation is convenient |
| Destination Image (Unstable Image) | DIU1. South Korea is a dangerous country.<br>DIU2. South Korea is a country where the war can occur at any time.<br>DIU3. The political situation and social security of South Korea are unstable |
| Tourism Attitude Change | TAC1. I would like to change the tourism destination and acquire more information about tourist attractions in other countries and regions because of the THAAD incident.<br>TAC2. I have decided to change the tourism destination and visit other countries and regions because of the THAAD incident. |
| Tourism Behavioral Intention | TBI1. I really want to visit South Korea even though it might be dangerous.<br>TBI2. I really want to visit South Korea if I have enough money even though it might be dangerous.<br>TBI3. I will visit many regions of South Korea and experience different feelings even though it might be dangerous.<br>TBI4. Traveling to South Korea is still the first choice for tourist destinations even though it might be dangerous. |

## 3.3. Statistical Analysis

First, we used exploratory factor analysis (EFA) to identify the underlying relationships between measured variables. EFA assumes that any measured variable may be associated with any factor. EFA is essential to determine underlying factors/constructs for a set of measured variables. Afterward, we moved on to confirmatory factor analysis (CFA). CFA was used to test the hypothesis that a relationship between the observed variables and their underlying latent factor(s)/construct(s) exists.

Second, convergent and discriminant validity were used to assess the construct of this study. Convergent validity refers to the degree to which two measures of constructs are related. Convergent validity can be estimated using correlation coefficients. Discriminant validity tests whether concepts that are not supposed to be related are actually unrelated.

At last, a structural equation model (SEM) was used to examine the theoretical model proposed above by using several indices, such as $\chi^2$/df, goodness of fit index (GFI), adjusted goodness of fit index (AGFI), normative fit index (NFI), comparative fit index (CFI), root mean square residual (RMR), root mean square error of approximation (RMSEA) and other fitting indices.

We used the SPSS 21.0 statistic program to perform frequency analysis, convergent validity, and discriminant validity. The CFA and SEM were carried out in AMOS 21.0.

## 4. Results

### 4.1. Demographic Characteristics

The demographic characteristics of respondents are shown in Table 2. The percentages of male and female were approximate. Unmarried ones accounted for 51.1% of total samples. Most of the respondents were between the ages of 20 and 60. The level of education generally concentrated in Junior colleges and above. The monthly income was mostly between 5000 and 10,000 CNY. Respondents' occupations covered a wide range and came from different industries of society.

**Table 2.** Demographic characteristics of respondents.

| Item | Category | Frequency | Item | Category | Frequency |
|---|---|---|---|---|---|
| Gender | Male | 473 (48.8%) | Monthly Income | Unde 5000 CNY | 233 (24.0%) |
| | Female | 497 (51.2%) | | 5000–10,000 CNY | 379 (39.1%) |
| Marital Status | Married | 465 (47.9%) | | 10,000–15,000 CNY | 229 (23.6%) |
| | Unmarried | 496 (51.1%) | | 15,000–20,000 CNY | 96 (9.9%) |
| | Others | 9 (0.9%) | | Above 20,000 CNY | 33 (3.4%) |
| Age | less than 19 | 53 (5.5%) | Occupation | Government administrative staff | 58 (6.0%) |
| | 20–29 | 256 (26.4%) | | Company's employer | 86 (8.9%) |
| | 30–39 | 295 (30.4%) | | Service industry | 81 (8.4%) |
| | 40–49 | 189 (19.5%) | | Private enterprise owners | 84 (8.7%) |
| | 50–59 | 163 (16.8%) | | Educator | 72 (7.4%) |
| | Above 60 | 14 (1.4%) | | Scientist | 66 (6.8%) |
| Level of Education | Senior high school | 135 (13.9%) | | Technical experts | 114 (11.8%) |
| | Junior college | 251 (25.9%) | | Medical staff and legal staff | 65 (6.7%) |
| | Bachelor's degree | 383 (39.5%) | | Manager | 55 (5.7%) |
| | Master's degree | 140 (14.4%) | | Housewife | 107 (11.0%) |
| | Doctor's degree | 41 (4.2%) | | Student | 151 (15.6%) |
| | Others | 20 (2.1%) | | Others | 31 (3.2%) |
| | Total | 970 | | Total | 970 |

## 4.2. Exploratory Factor Analysis

At first, Cronbach's $\alpha$ was used to estimate the reliability of the whole sample data. The results showed that the Cronbach's $\alpha$ of all variables (including tourism risk perception, destination image, and tourism attitude change, tourist behavioral intention) was larger than 0.8, indicating high intercorrelations among test items (Table 3). The Kaiser–Meyer–Olkin (KMO) value is a statistic that indicates the proportion of variance in variables that might be caused by underlying factors. The results showed that the KMO values of all variables were all greater than 0.7, indicating that factor analysis may be useful with the data. Bartlett's test of sphericity tests the hypothesis that whether variables are unrelated and, therefore unsuitable for structure detection. All the *p*-values of the Bartlett spherical test were less than 0.05, indicating that factor analysis may be useful with our data. The results of EFA showed that the cumulative variance of all variables was larger than 80% (Table 3), indicating that the factors extracted by EFA could to a large extent explain the variance of the data.

**Table 3.** Results of exploratory factor analysis.

| Factor (Reliability) | MeasurementItems | Factor Loading | Common Feature | Eigenvalue | Contribution |
|---|---|---|---|---|---|
| Tourism Risk Perception (0.935) | TRP1 | 0.923 | 0.852 | | |
| | TRP2 | 0.856 | 0.734 | 3.349 | 88.727 |
| | TRP3 | 0.946 | 0.896 | | |
| | TRP4 | 0.931 | 0.867 | | |
| Cumulative variance (%): 83.727%, KMO (Kaiser–Meyer–Olkin): 0.853, Bartlett test: 3515.559 (*p*-value: 0.000) | | | | | |
| Tourism Image (0.883) | DIT1 | 0.841 | 0.802 | | |
| | DIT2 | 0.579 | 0.861 | 3.117 | 23.975 |
| | DIT3 | 0.722 | 0.873 | | |
| | DIT4 | 0.750 | 0.795 | | |
| Culture Image (0.805) | DIC1 | 0.795 | 0.871 | 2.881 | 22.159 |
| | DIC2 | 0.611 | 0.747 | | |
| Stable Image (0.906) | DIS1 | 0.685 | 0.750 | | |
| | DIS2 | 0.748 | 0.809 | 2.769 | 21.297 |
| | DIS3 | 0.809 | 0.828 | | |
| | DIS4 | 0.809 | 0.822 | | |
| Unstable Image (0.950) | DIU1 | 0.956 | 0.916 | | |
| | DIU2 | 0.942 | 0.902 | 1.884 | 14.491 |
| | DIU3 | 0.950 | 0.916 | | |
| Cumulative variance (%): 81.922%, KMO (Kaiser–Meyer–Olkin): 0.911, Bartlett test: 10585.681 (*p*-value: 0.000) | | | | | |
| Tourism Attitude Change (0.954) | TAC1 | 0.978 | 0.957 | 1.914 | 95.704 |
| | TAC2 | 0.978 | 0.957 | | |
| Cumulative variance (%): 95.704%, KMO (Kaiser–Meyer–Olkin): 0.500, Bartlett test: 1746.418 (*p*-value: 0.000) | | | | | |
| Tourism Behavior Intension (0.965) | TBI1 | 0.947 | 0.897 | | |
| | TBI2 | 0.956 | 0.914 | 3.637 | 90.931 |
| | TBI3 | 0.952 | 0.907 | | |
| | TBI4 | 0.959 | 0.920 | | |
| Cumulative variance (%): 90.931%, KMO (Kaiser–Meyer–Olkin): 0.876, Bartlett 4918.267 (*p*-value: 0.000) | | | | | |

Note: All the abbreviations of variables are shown in Table 1.

### 4.3. Confirmatory Factor Analysis

After the EFA, we performed the CFA on the measurement model (Table 4). All the fit indices $\chi^2$ = 502.679, df = 196, GFI = 0.956, AGFI = 0.939, RMR = 0.030, NFI = 0.978, TLI = 0.982, CFI = 0.986, (RMSEA = 0.040) showed satisfactory levels, suggesting that the data fit the measurement model.

**Table 4.** Result of confirmatory factor analysis.

| Factor and Measured Index | | Estimate | Standardized Estimate | Standard Error | T Statistics |
|---|---|---|---|---|---|
| Tourism Risk Perception | TRP1 | 1.000 | 0.921 | | |
| | TRP2 | 1.036 | 0.943 | 0.020 | 52.415 *** |
| | TRP3 | 0.793 | 0.777 | 0.024 | 32.950*** |
| | TRP4 | 1.072 | 0.937 | 0.030 | 36.112 *** |
| Tourism Behavioral Intention | TBI1 | 1.000 | 0.958 | | |
| | TBI2 | 0.899 | 0.924 | 0.016 | 57.758 *** |
| | TBI3 | 0.895 | 0.932 | 0.015 | 59.912 *** |
| | TBI4 | 0.797 | 0.932 | 0.015 | 53.364 *** |
| Tourism Attitude Change | TAC1 | 1.021 | 0.941 | 0.026 | 38.700 *** |
| | TAC2 | 1.000 | 0.971 | | |
| Tourism Image | DIT1 | 1.000 | 0.845 | | |
| | DIT2 | 1.022 | 0.879 | 0.034 | 29.770 *** |
| | DIT3 | 1.147 | 0.879 | 0.034 | 33.668 *** |
| | DIT4 | 0.784 | 0.751 | 0.029 | 27.489 *** |
| Culture Image | DIC1 | 1.000 | 0.820 | | |
| | DIC2 | 1.074 | 0.821 | 0.036 | 29.853 *** |
| Stable Image | DIS1 | 1.000 | 0.820 | | |
| | DIS2 | 0.976 | 0.822 | 0.028 | 34.530 *** |
| | DIS3 | 1.169 | 0.874 | 0.036 | 32.612*** |
| | DIS4 | 0.997 | 0.842 | 0.032 | 31.509 *** |
| Unstable Image | DIU1 | 1.000 | 0.936 | | |
| | DIU2 | 1.042 | 0.918 | 0.020 | 51.444 *** |
| | DIU3 | 0.958 | 0.936 | 0.018 | 54.597 *** |

Note: All the abbreviations of variables are shown in Table 1. *** $p < 0.01$.

### 4.4. Convergent and Discriminant Validity

Table 5 presents the convergent and discriminant validity statistics. All of the average variance extracted (AVE) and composite reliability (C.R.) values for the multi-item scales were greater than the minimum levels of 0.5 and 0.7, respectively [39], indicating a sufficient level of convergent validity for the measurement model. Concerning the discriminant validity of the constructs, the square root of AVE of each construct was greater than the correlation coefficients for the corresponding inter-constructs in most cases, which confirmed the discriminant validity of the constructs [40].

**Table 5.** The correlation matrix of the measurement model.

| Potential Variables | TRP | DIT | DIC | DIS | DIU | TAC | TBI |
|---|---|---|---|---|---|---|---|
| TRP | 1 | | | | | | |
| DIT | −0.354 *** | 1 | | | | | |
| DIC | −0.405 *** | 0.785 *** | 1 | | | | |
| DIS | −0.524 *** | 0.767 *** | 0.769 *** | 1 | | | |
| DIU | 0.492 *** | −0.113 *** | −0.157 *** | −0.207 *** | 1 | | |
| TAC | 0.301 *** | −0.198 *** | −0.272 *** | −0.302 *** | 0.199 *** | 1 | |
| TBI | −0.193 *** | 0.282 *** | 0.349 *** | 0.402 *** | 0.023 | −0.627 *** | 1 |
| Average | 2.698 | 3.940 | 3.777 | 3.877 | 2.625 | 3.197 | 2.850 |
| SD | 1.101 | 0.737 | 0.933 | 0.825 | 1.089 | 0.869 | 1.280 |
| C.R. | 0.914 | 0.931 | 0.798 | 0.919 | 0.936 | 0.741 | 0.945 |
| AVE | 0.717 | 0.772 | 0.664 | 0.739 | 0.831 | 0.536 | 0.810 |
| √AVE | 0.847 | 0.879 | 0.815 | 0.860 | 0.912 | 0.732 | 0.900 |

Note: All the abbreviations of variables are shown in Table 1. SD = standard deviation, C.R. = composite reliability, AVE = average variance extracted. *** $p < 0.01$.

### 4.5. Structural Equation Model

Table 6 and Figure 2 summarizes the estimated results of the proposed research model. Results showed that all the fit indices except RMR were within the range of the general fitness index (Table 6). Therefore, the data was suitable for using SEM. The standardized path coefficients of SEM are shown in Figure 2. First, tourism risk perception had significant negative effects on tourism image (path

coefficient = −0.404, *p* < 0.01), cultural image (path coefficient = −0.478, *p* < 0.01), stable image (path coefficient = −0.573, *p* < 0.01), but had significant positive effect on the image of instability (path coefficient = 0.537, *p* < 0.01). Therefore, hypothesis 1-1 to 1-4 were supported by this study. Second, tourism risk perception had a significant positive effect on tourism attitude change (path coefficient = 0.313, *p* < 0.01). Therefore, hypothesis 2 was supported. Third, tourism risk perception had a significant negative effect on tourism willingness (path coefficient = −0.154, *p* < 0.01), so hypothesis 3 was also supported (Table 7).

The squared multiple correlations (SMC) can reflect the explanatory variance of each variable to tourism risk perception. The results showed that the destination image has the highest explanatory variance (16.3% for tourism image, 22.9% for the cultural image, 32.9% for the stable image, 28.9% for the unstable image), followed by tourism attitude change (9.8%) and tourist behavioral intention (2.4%).

**Table 6.** The goodness of fit for the structural equation model.

| Index | $\chi^2$ | df | *p* | Normed $\chi^2$ | GFI | AGFI | RMR | NFI | TLI | CFI | RMSEA |
|---|---|---|---|---|---|---|---|---|---|---|---|
| Value | 663.466 | 223 | 0.000 | 2.975 | 0.947 | 0.929 | 0.119 | 0.971 | 0.976 | 0.981 | 0.045 |
| Suggested | | | | 1-3 | ≥0.90 | ≥0.90 | <0.05 | ≥0.90 | ≥0.90 | ≥0.90 | <0.05 |

**Table 7.** Verification result of all hypotheses proposed in this study.

| | Hypothesis | Support/Not |
|---|---|---|
| **H1** | Tourism risk perception has a significant impact on destination image. | Supported |
| **H1-1** | Tourism risk perception has a significant negative impact on the tourism image. | Supported |
| **H1-2** | Tourism risk perception has a significant negative impact on the cultural image. | Supported |
| **H1-3** | Tourism risk perception has a significant negative impact on the stability image. | Supported |
| **H1-4** | Tourism risk perception has a significant positive impact on the instability image. | Supported |
| **H2** | Tourism risk perception has a significant positive impact on tourism attitude change. | Supported |
| **H3** | Tourism risk perception has a significant negative impact on tourist behavioral. intention. | Supported |

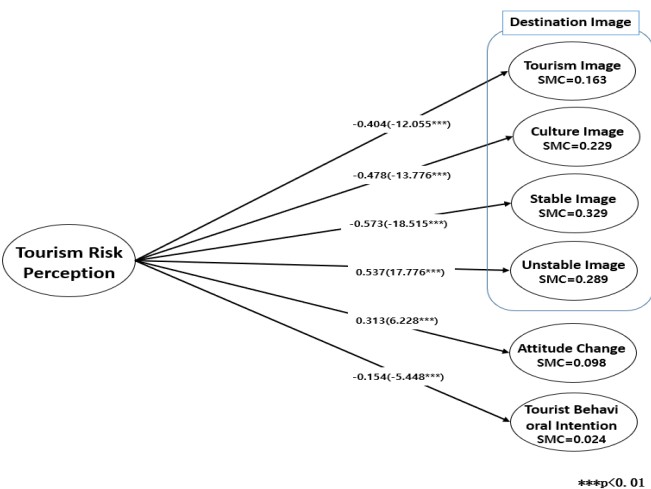

**Figure 2.** The results of the structural equation model. The standardized path coefficients (*t*-statistic in bracket) and the squared multiple correlations (SMC) are shown.

## 5. Discussion and Conclusions

### 5.1. Discussion

5.1.1. Comparison with Previous Studies

This study showed that there is a close relationship between Chinese citizens' risk perception and the image of the tourist destination, which supported the findings of previous studies [32,33]. For example, Li [32] used the questionnaire method to study the risk perception of tourists visiting

Tibet, China and found that the locations where tourists perceived a higher risk had a worse destination image and fewer tourists. Due to the impact of frequent earthquakes and the risk of radiation exposure through contaminated food, water, and air quality, socio-psychological, and financial risks perceived by tourists influenced destination images of Japan [33]. Thus, crisis events such as natural disasters, political turbulence, and social instability will increase the degree of tourism risk perception and thus affect the image of tourist destinations.

We also found that tourism risk perception has a significant positive effect on tourism attitude change. This result supported the findings of several previous studies [4,34–36]. Sönmez and Graefe [34] found that past travel experience of international travelers to specific regions increases the intention to travel there again but decreases the intention to risky areas due to perceived risks. The majority of international travelers were more likely to change their travel plans to a destination that has elevated risk, while the minority reports they were more unlikely [35]. Similarly, for tourists planning to South Korea perceived the risks caused by the THAAD incident, some of them would change their original destinations, while others would choose to suspend their tourism activities.

Furthermore, after the THAAD incident, Chinese tourists' concerns over the political situation between China and South Korea had led to a significant change in the tourist behavioral intention. This result supported the findings of several previous studies [11,37]. For example, Cater [37] found that the most successful adventure tourism operators in Queenstown, New Zealand, were those that had reduced their actual risk levels. When analyzing a theoretical framework of vacation decisions involving terrorism risk, Sönmez and Graefe [34] found that international attitude, risk perception level, and income were found to directly influence international vacation destination choice. Thus, our results and these studies suggested that tourism risk perception has a significant negative effect on tourist behavioral intention.

### 5.1.2. Theoretical Contributions and Implications for Sustainable Tourism

In this study, in the context of THAAD incident, by combing the theoretical concepts of tourism risk perception, destination image, tourism attitude change, and behavioral intent, a structural equation model was developed and the causal relationships between various conceptual variables were clarified, which promoted the theoretical development of tourism risk perception. In addition, most of the previous studies about the impacts of crisis events on tourism focused on natural events and social events [8,10,14,34], but this study focused on a political event, which filled the gap in the impact of the political crisis of the two countries on the tourism industry and associated economic and social development. Therefore, this study enriched the diversity of research objects in tourism studies. Our results suggested that if any country wants to develop international tourism, it must maintain social and political stability and a friendly attitude towards neighboring countries. The international tourism industry can develop well only in the condition of good bilateral relations.

The results of this study indicated that political crisis events could affect the sustainable development of society and economy through tourists' risk perception. After the THAAD incident, the destination image of Chinese tourists on South Korea suddenly decreased, causing negative emotions and consumer animosity toward South Korea and large-scale cancellation or replacement of tourist destinations. Without the huge Chinese tourism market, the Korean tourism industry would experience a long-term downturn, which will not only lead to the high unemployment rate and the operating difficulties of major scenic spots and travel agencies but also affect the transportation, catering, accommodation, and other industries. Thus, the THAAD incident has a negative impact on the sustainable development of society and economy.

How to maintain the sustainable development of culture, biodiversity, and life support systems is the core of sustainable tourism [41]. Before the THAAD incident, a large part of Chinese tourists traveled to the important historical and cultural heritage sites (such as Hahoe and Yangdong, both of which are World Cultural Heritage Sites) in South Korea [15]. Due to the long history of these heritage sites, the cost of upkeep and maintenance is very high and requires sufficient tourism income to

support them. Thus, tourism consumption behavior has a positive effect on the protection of related cultural heritage. However, after the THAAD incident, due to the sudden decline in the number of tourists, the revenue of the local government and the tourism management department had decreased, which was not conducive to the sustainable development of tourism. Therefore, we need to actively cope with the negative impact of the crisis events on tourism.

### 5.1.3. Practical Recommendations

How to deal with the negative impact of crisis events on tourism destinations and surrounding areas needs to combine crisis-related theories with practice. After the crisis, how the tourism industry can recover from the effects of the crisis is an important issue for tourism managers. We have the following practical recommendations:

First, for the South Korea government, especially tourism management departments, they should simplify visa approval procedures and procedures, and increase the number of flights to cities in central and western China and cruise lines in southeast coastal port cities. Meanwhile, they should increase opportunities for mutual exchanges to resolve contradictions and conflicts, and promote friendship between the two countries.

Second, tourism destination management agencies in Korea should actively engage in cooperation with Chinese tourism enterprises such as organizing publicity exhibitions and inviting Chinese tourism companies to South Korea for investigation and communication, so as to improve the destination image. The introduction of new services such as the Chinese language guide is also useful to increase the willingness of Chinese tourists to visit and to recommend Korea to others.

Third, for tourism companies, they could attract more Chinese tourists by continuing to introduce attractive and competitive tourism routes and launch low-cost promotional marketing policies.

### 5.1.4. Limitations and Further Research Direction

There may be some possible limitations in this study. As this study started at the sensitive period of the THAAD incident, Chinese citizens traveling to South Korea declined sharply. Thus, we mainly focused on the potential tourism market rather than exploring the difference between the existing and potential tourism market. Thus, further research should compare the tourism risk perception between Chinese people who planned to South Koreas and who have traveled there before.

Although the survey sampling in this study followed objective and comprehensive standards, including Chinese citizens from different regions, ages, genders, and occupations who have the economic ability and objective conditions to travel to South Korea. However, the sample size was rather limited, which may cause the sampling error. Further studies could enlarge the sample size of respondents, so as to compare the differences in tourism attitude and tourism behavior intention among different periods and regions. This effort could provide a more precise marketing strategy and policy guidance for the tourism management department.

### 5.2. Conclusions

In this study, a theoretical model is constructed to analyze the effect of THAAD Incident on tourism risk perception, destination image, attitude change, and tourist behavioral intention of Chinese citizens traveling to Korea. Based on the questionnaire survey of 1000 Chinese citizens from 34 provinces of China, we used various statistical methods to verify the theoretical model. The following conclusions could be drawn:

(1)　There is a close relationship between Chinese citizens' risk perception and the image of the tourist destination. The strong risk perception of the destination would destroy the tourism image, cultural image, and stable image, and enhance the unstable image of the destination.

(2)    Tourism risk perception has a significant positive effect on tourism attitude change. Stronger risk perception of the tourism destination would promote the change in tourism attitude. Chinese citizens who plan to travel to South Korea tend to change their destinations.

(3)    Tourism risk perception has a significant negative effect on tourist behavioral intention. Stronger risk perception of the tourism destination would reduce Chinese citizens' intention to travel to South Korea and even make them cancel their travel plans.

Overall, the conclusions of this study have important practical significances for the sustainable development of tourism in China and South Korea after the THAAD incident and can provide a theoretical basis for formulating and improving tourism risk management strategies.

**Author Contributions:** Data curation, H.Z.; formal analysis, H.Z.; methodology, T.C.; project administration, H.W.; software, T.C.; writing—original draft, H.Z.; writing—review and editing, H.W. All authors have read and agreed to the published version of the manuscript.

**Funding:** This research was funded by the project of the Chinese Academy of Sciences (grant number: KGFZD-135-17-009-1).

**Acknowledgments:** We are grateful to all our interview partners for spending their time and efforts on answering our questions. We thank three external reviewers who helped to significantly improve this paper.

**Conflicts of Interest:** The authors declare no conflict of interest.

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
