# Peer review of "The Impact of a Terminal High Altitude Area Defense Incident on Tourism Risk Perception and Attitude Change of Chinese Tourists Traveling to South Korea"

_sustainability, doi:10.3390/su12010007_

Round 1

Reviewer 1 Report

Thank you for the opportunity of reviewing your interesting article. It addresses an interesting topic and it is well prepared and well drafted.

However, the literature review part is quite limited for an article of this extension and little literature on the topic is actually referenced. Consequently, this part should be enhanced.

The research design is appropriate and the results are adequately presented.

In the final pat there is no reference to the managerial/policy implications. I thing the paper should be improved by adding reference to the utility of the findings. Moreover, addressing the limitations and providing some further directions for research would be beneficial.

Good luck with the review!

Author Response

the literature review part is quite limited for an article of this extension and little literature on the topic is actually referenced. Consequently, this part should be enhanced. In the final part there is no reference to the managerial/policy implications. I think the paper should be improved by adding reference to the utility of the findings.

Response: Thank you for your comments. We have added more 10 references about the topic, especially in the Introduction and Discussion section.

Moreover, addressing the limitations and providing some further directions for research would be beneficial.

Response: In the revised paper, we add section 5.14 to show further directions for research.

“There may be some possible limitations in this study. As this study started at the sensitive period of THAAD Incident, Chinese citizens traveling to South Korea declined sharply. Thus, we mainly focused on the potential tourism market rather than exploring the difference between the existing and potential tourism market. Thus, further research should compare the tourism risk perception between Chinese people who planned to South Koreas and who have traveled there before.

Although the survey sampling in this study followed objective and comprehensive standards, including Chinese citizens from different regions, ages, genders, occupations who have the economic ability and objective conditions to travel to South Korea. However, the sample size was rather limited, which may cause the sampling error. Further studies could enlarge the sample size of respondents, so as to compare the differences in tourism attitude and tourism behavior intention among different periods and regions. This effort could provide a more precise marketing strategy and policy guidance for the tourism management department.”

Reviewer 2 Report

This paper provides interesting information on tourism risk perception and the effect it has on visitors' behavior and destination image in the context of the THAAD incident in South Korea. Although it is well written and it provides original information on the studies theme, it needs several improvements. First of all it is too focused on the analyzed context and it does not overview how the used concepts may be generalized in similar situations. What is the theoretical value that this paper brings to the tourist literature? Also, considering the specificities of the Sustainability journal, I do not think that this paper is much focused on sustainability issues. In the end of abstract, authors mention that: "This study is of great importance for the sustainable development of tourism", however, the sustainable tourism concept is not tackled in the paper. In the introduction section, authors say: "The number of outbound tourists reached 12238 million, and the consumption exceeded 108.9 billion US$." They should mention the referenced year when these results were registered in China's tourism indicators. Also, recent statistics should be mentioned. Moreover, the introduction is only focused on describing the practical context of the paper, without clearing out the conceptual framework and relevant variables of the study. The scope and objectives of the study should be highlighted in this section with a more specific focus on the academic literature. In the second section of the paper, different theoretical concepts are presented in a separate way. They should be interconnected and authors should provide linkages between these concepts and paragraphs. In the research design section, authors should mention how they have chosen the items included in the questionnaire and refer to previous studies which might have used similar measurement items in different contexts. In the discussion section, the authors should provide information on the theoretical contributions that their paper brings to the existing literature. What is the original feature of their study? Also, they should mention whether their model and their results confirm or not previous studies on the same subject. All these sources should be mentioned for each variable and relationship presented in the theoretical model. Furthermore, only one practical recommendation is proposed in this section for tourism management departments and related companies in China and South Korea. What about Korea’s destination management organizations? Other practical recommendations should be proposed and described. Finally, the reference list is quite short and it does not include sufficient recent studies.

Author Response

First of all it is too focused on the analyzed context and it does not overview how the used concepts may be generalized in similar situationWhat is the theoretical value that this paper brings to the tourist literature? Also, considering the specificities of the Sustainability journal, I do not think that this paper is much focused on sustainability issues. In the end of abstract, authors mention that: "This study is of great importance for the sustainable development of tourism", however, the sustainable tourism concept is not tackled in the paper.

Thank you for your comments. We add some information about sustainable tourism in the revised paper (Section 5.1.2):

“How to maintain the sustainable development of culture, biodiversity, and life support systems is the core of sustainable tourism [44]. Before the THAAD incident, a large part of Chinese tourists traveled to the important historical and cultural heritage sites (such as Hahoe and Yangdong, both of which are World Cultural Heritage Sites) in South Korea [15]. Due to the long history of these heritage sites, the cost of upkeep and maintenance is very high and requires sufficient tourism income to support them. Thus, tourism consumption behavior has a positive effect on the protection of related cultural heritage. However, after the THAAD incident, due to the sudden decline in the number of tourists, the revenue of the local government and the tourism management department had decreased, which was not conducive to the sustainable development of tourism. Therefore, we need to actively cope with the negative impact of the crisis events on tourism.”

In the introduction section, authors say: "The number of outbound tourists reached 12238 million, and the consumption exceeded 109.8 billion US$." They should mention the referenced year when these results were registered in China's tourism indicators. Also, recent statistics should be mentioned.

Response: We added the referenced year in the revised paper:

“By 2016, 8.268 million Chinese citizens had traveled to Korea, occupying 47.5% of the total number of Korean inbound tourists [16].”

Moreover, the introduction is only focused on describing the practical context of the paper, without clearing out the conceptual framework and relevant variables of the study. The scope and objectives of the study should be highlighted in this section with a more specific focus on the academic literature.

Response: we totally rewrite the Introduction Section and add the conceptual framework and relevant variables of the study. The scope and objectives of this study were also highlighted. Please find it in the revised paper.

In the second section of the paper, different theoretical concepts are presented in a separate way. They should be interconnected and authors should provide linkages between these concepts and paragraphs.

Response: In section 2.2.2, we provide linkages between these concepts. Please find it in the revised paper.

In the research design section, authors should mention how they have chosen the items included in the questionnaire and refer to previous studies which might have used similar measurement items in different contexts.

Response: We refer to the previous studies about how to choose the items:

“The measurement index of tourism risk perception [34], destination image [26], tourism attitude change [37] and tourism behavior intention [35] were obtained from corresponding references.”

In the discussion section, the authors should provide information on the theoretical contributions that their paper brings to the existing literature. What is the original feature of their study? Also, they should mention whether their model and their results confirm or not previous studies on the same subject. All these sources should be mentioned for each variable and relationship presented in the theoretical model.

Response: The theoretical contributions of this study are added in Section 5.1.2 and whether our model and results confirm or not previous studies on the same subject were added in Section 5.1.1.

Furthermore, only one practical recommendation is proposed in this section for tourism management departments and related companies in China and South Korea. What about Korea’s destination management organizations? Other practical recommendations should be proposed and described.

Response: We add several recommendations in Section 5.1.3 for tourism management departments and Korea’s destination management organizations.

Finally, the reference list is quite short and it does not include sufficient recent studies.

Response: In the revised paper, we have added more 10 references for the topic.

Reviewer 3 Report

The topic of research is of great relevance. The article is well structured, follows the typical structure of an SEM model. It is easy to read. The bibliography is insufficient. However, the content is very short and has great limitations.

My comments and suggestions are:

1.-  The title is very long and it is also not convenient to introduce acronyms in the same one that the reader does not know (THAAD). Reduce it.

2.-  In the introduction section:

The subject under study is not properly contextualized. There are many questions that should be answered in the introduction: What other studies exist on this subject? What have you discovered? What is the gap that is identified in the literature, taking these studies into account? What is the novelty of the study? Who are the results valid for?

Briefly refer to the methodology, population and sample that is used to meet the objectives after the objectives.

Include a final paragraph that refers to the sections in which the paper is divided.

3.-  Literature review section: the variables are defined very briefly. It should be completed.
The main limitation is that they do not carry out an adequate review of the literature on which to rely to define each of the proposed causal relationships. One of the basic premises for proposing a theoretical model to estimate is that the hypotheses support each of them in the previous literature. Complete this review.
4.- In the model of figure 1 not all the hypotheses are reflected (H1.1, H1.2, H1.3, H1.4), to propose these hypotheses no literature review is performed. It is necessary and essential to review the literature to propose these hypotheses.
5.- The questionnaire of the study of Where was it obtained? cite if it is own elaboration or are scales of other studies.
6.- What is the sample error of the study?
7.- Perform the Harman test to analyze common bias.
8.- Discussion section: it is practically non-existent. A discussion of each of the results of the proposed causal relationships should be provided taking into account whether these results are corroborated by other studies or not. Properly complete this section.
9.- In the discussion, mention the limitations of the study.

10.- Bibliography: 25 references is clearly insufficient for a topic already studied in the previous literature.

Author Response

1.-  The title is very long and it is also not convenient to introduce acronyms in the same one that the reader does not know (THAAD). Reduce it.

Response: We revised the title and introduce acronyms of THAAD in the new title. The revised title is: The impact of Terminal High Altitude Area Defense incident on tourism risk perception and attitude change of Chinese tourist traveling to South Korea

2.-  In the introduction section:

The subject under study is not properly contextualized. There are many questions that should be answered in the introduction: What other studies exist on this subject? What have you discovered? What is the gap that is identified in the literature, taking these studies into account? What is the novelty of the study? Who are the results valid for? Briefly refer to the methodology, population and sample that is used to meet the objectives after the objectives. Include a final paragraph that refers to the sections in which the paper is divided.

Response: Thank you for your comments. We agree that the original Introduction Section was too short. In the revised paper, we totally re-wrote the Introduction section. In the new Introduction, we briefly place the study in a broad context and highlight why it is important. We also define the purpose of the work and its significance. We also briefly mention the main aim of the work. At last, we showed the sections in which the paper is divided. We hope that there revisions could meet your requirements.

3.-  Literature review section: the variables are defined very briefly. It should be completed.
The main limitation is that they do not carry out an adequate review of the literature on which to rely to define each of the proposed causal relationships. One of the basic premises for proposing a theoretical model to estimate is that the hypotheses support each of them in the previous literature. Complete this review.

Response: We add many new studies to define each variable, and proposing a theoretical model based on the previous literature. Please find the completed review in Section 2. Literature Review, Research Hypotheses, and Model.

4.- In the model of figure 1 not all the hypotheses are reflected (H1.1, H1.2, H1.3, H1.4), to propose these hypotheses no literature review is performed. It is necessary and essential to review the literature to propose these hypotheses.

Response: Following your comments, We revised figure 1 to show all the hypotheses, and reviewed the literature to propose these hypotheses.

5.- The questionnaire of the study of Where was it obtained? cite if it is own elaboration or are scales of other studies.

Response: We refer to the previous studies about how to choose the items:

“The measurement index of tourism risk perception [34], destination image [26], tourism attitude change [37] and tourism behavior intention [35] were obtained from corresponding references.”

6.- What is the sample error of the study?

Response: Although the survey sampling in this study followed objective and comprehensive standards, the sample size was rather limited, which may could not represent the overall structure of Chinese tourists. This may cause the sampling error. This is a limitation of this study. We discussed this issue in Section 5.1.4.

7.- Perform the Harman test to analyze common bias.

Harman’s Single-Factor Approach (Podsakoff and Organ, 1986) was used to control for common method bias (CMB). This approach can be assessed by both EFA (exploratory factor analysis) and CFA (Podsakoff et al., 2003). Therefore, in studies using EFA and CFA, the Harman test was not performed (such as the following references). Following these studies, we did not perform the Harman test in this study.

[1] Dedeoglu B B, Bilgihan A, Ye B H et al. The Impact of Servicescape On Hedonic Value and Behavioral Intentions: The Importance of Previous Experience[J]. International Journal of Hospitality Management, 2018, 72: 10-20.

[2] Zhu H, Liu J, Wei Z, et al. Residents’ Attitudes towards Sustainable Tourism Development in a Historical-Cultural Village: Influence of Perceived Impacts, Sense of Place and Tourism Development Potential. Sustainability, 2017, 9(1}, 61.

[3] Song H, You G, Reisinger Y, et al. Behavioral intention of visitors to an Oriental medicine festival: An extended model of goal directed behavior. Tourism Management, 2014, 42: 101-113.

8.- Discussion section: it is practically non-existent. A discussion of each of the results of the proposed causal relationships should be provided taking into account whether these results are corroborated by other studies or not. Properly complete this section.

Response: We add a discussion of each of the results of the proposed causal relationships in Section 5.1.1

9.- In the discussion, mention the limitations of the study.

Response: we add the limitations of the study in Section 5.1.4.

10.- Bibliography: 25 references is clearly insufficient for a topic already studied in the previous literature.

Response: In the revised paper, we have added more 10 references for the topic.

Round 2

Reviewer 2 Report

The paper has been improved and may be considered for publication in the Sustainability Journal.

Reviewer 3 Report

The authors responded adequately to each of the recommendations and suggestions. The paper in the current state is very interesting and presents a high level of scientific quality. I congratulate the authors for the work done.